# Ecological Footprint as an Indicator of Corporate Environmental Performance—Empirical Evidence from Hungarian SMEs

Áron Szennay [1,2], Cecília Szigeti [3,*], Judit Beke [4] and László Radácsi [5]

1   Department of Finance, Faculty of Finance and Accountancy, Budapest Business School,
    1149 Budapest, Hungary; szennay.aron@uni-bge.hu
2   Doctoral School of Regional and Economic Sciences, Széchenyi István University, 9026 Győr, Hungary
3   Department of International and Theoretical Economics, Kautz Gyula Economics Faculty, Széchenyi István
    University, 9026 Győr, Hungary
4   Department of International Economics, Faculty of International Management and Business,
    Budapest Business School, 1165 Budapest, Hungary; lisanyi.endrene@uni-bge.hu
5   Department of Management, Faculty of Finance and Accountancy, Budapest Business School,
    1149 Budapest, Hungary; radacsi.laszlo@uni-bge.hu
*   Correspondence: szigetic@sze.hu

**Abstract:** Small- and medium-sized enterprises (SMEs) play a significant role in the national economies of the EU member states. This economic activity has an inevitable environmental impact; however, environmental performance indicators are mostly measured at larger companies. Since the ecological footprint (EF) is a suitable measure of unsustainability, this paper considers it as a measure of the environmental impact of SMEs. An EF calculator for SMEs was developed that is freely available online, and it is a methodological innovation per se. Our previous research projects highlighted that the calculator must be easy-to-use and reliable; therefore, the calculator considers only the common, standardizable, and comparable elements of EF. Our results are based on validated ecological footprint data of 73 Hungarian SMEs surveyed by an online ecological footprint calculator. In order to validate and test the usefulness of the calculator, interviews were conducted with respondents, and results were also checked. The paper presents benchmark data of ecological footprint indicators of SMEs obtained from five groups of enterprises (construction, white-collar jobs, production, retail and/or wholesale trade, and transportation). Statistical results are explained with qualitative data (such as environmental protection initiatives, business models, etc.) of the SMEs surveyed. Our findings could be used as a benchmark for the assessment of environmental performance of SMEs in Central- and Eastern Europe.

**Keywords:** ecological footprint; environmental performance of SMEs

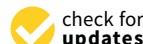



## 1. Introduction

There is a broad consensus around the need and usefulness of indicators and metrics to define the planetary boundaries. Humanity's demand on resources has been expanding, which has a significant impact on the Earth system; therefore, many researchers now believe that this era can be considered as a new geological epoch, the so-called Anthropocene [1]. The World Overshoot Day, calculated by Global Footprint Network (GFN), is a high-level and easy-to-understand indicator of global (un)sustainability, since it "marks the date when humanity's demand for ecological resources and services in a given year exceeds what Earth can regenerate in that year" [2]. Since 1970, this date occurs before 31st December each year, and, since the beginning of the 2010s, it lands around the 1st of August. This figure means that in the 2010s, humanity used up approximately 1.7 times more resources each year than the ecosystems of the Earth can regenerate. Although environmentally friendly (i.e., "green") consumption habits and technologies are becoming more common, recent studies show that even conscious consumers change their habits occasionally (e.g.,

during holiday) [3]. Considering this, it is not surprising that Mathis Wackernagel [4] called our economy the largest Ponzi scheme ever. However, as a result of the COVID-19 pandemic and the lockdown measures introduced in the developed and emerging world, the Overshoot Day landed on 22nd August in 2020.

Environmental sustainability (with special regard to the reduction of greenhouse gas emission and the increase of renewable energies) is one of the headline targets the Europe 2020 strategy of the European Union (EU) [5]. Since Europe's 25 million small- and medium-sized enterprises (SMEs) play a significant role in the economies of EU member states, their contribution to sustainable development is also crucial. SMEs make up over 99% of all enterprises in all EU countries, they generate around two-thirds of all jobs and account for more than half of EU's GDP [6]. Evidence shows that both the regulatory stakeholder pressure and organizational stakeholder pressure positively influence green production practices, corporate reputation, and the environmental performance of manufacturing SMEs [7], which means that the sustainability efforts of both the EU as a whole and the individual member states have a positive impact on the attitudes of SME managers towards sustainability. This finding is supported by evidence from the energy sector, i.e., debt increases the value of SMEs in countries with strong environmental commitment, which makes it possible to facilitate growth with additional external capital [8]. Italian evidence highlights, however, that decision-makers of SMEs "have a high school diploma mainly used bank loans or overdrafts as compared to those that received formal training" [9]. Nonetheless, firms with external capital must maintain financial capacity to repay it, which might create significant problems in case of a crisis situation [10], and capital structure considerations may also play a crucial role [9,11,12]. Another aspect is that a large share of SMEs are family businesses that make up between 57 and 66 percent of the enterprises with 3 and 99 employees in Hungary [13]. Evidence shows that Hungarian family businesses have better chances of survival and create higher value added than non-family businesses [14].

Experience has shown that, although several managers of SMEs are interested in metrics on environmental performance, their businesses/companies cannot afford paying for comprehensive environmental audit and advisory; therefore, they do not have enough experience in selecting the most appropriate measures. Our results suggest that the ecological footprint (EF) is a suitable metric for SMEs because (1) it is easy to understand, therefore making it easy for even managers who do not have enough relevant expertise to use it; (2) the calculation is standardizable, therefore capable of providing performance metrics at a low cost or even for free; and (3) quantitative performance indicators allow them to support the selection of the most appropriate projects or measures to enhance corporate environmental performance (CEP). Our aim was to develop an easy-to-use EF calculator for SMEs which could measure the common elements of corporate environmental impacts reliably. Based on experiences with carbon footprint calculators, it has been found that there is a trade-off between accuracy and simplicity. A calculator that measures the EF of SMEs is needed because, whereas large enterprises have sufficient resources to make unique calculations, it can be difficult for SMEs to find resources and expertise [15]. The results of standardized calculations can be complemented with unique items (e.g., material consumption or more sophisticated data on meals) or longitudinal assessment of CEP can be conducted. Based on the results of our previous analyses, the usefulness and accuracy of the calculator developed was validated, the results were discussed with the respondents, and we made attempts to improve the calculator [16]. Nevertheless, lacking benchmark data can be considered as the most critical problem. Therefore, this paper aimed to calculate sectoral comparative benchmark data.

This paper is structured as follows: Chapter 2 summarizes the concept of the EF and its potential role in measuring of corporate environmental performance. At the end of the chapter, some examples of sectoral EF calculations are presented. The third chapter gives an account of the methodology and the sample used, while the fourth chapter summarizes our results.

## 2. Theoretical Framework

### 2.1. The Concept of Ecological Footprint

The ecological footprint (EF) concept was developed by Mathis Wackernagel and William E. Rees [17] in 1996. Since the introduction of the concept, the EF has been used to measure environmental sustainability both at a global level and of individual consumption, as well [18–22]. Nonetheless, other indicators could also be used for measuring environmental sustainability [23–25], but it is only the EF that indicates the upper limit of growth properly [26]. The GFN started its National Footprint Accounts (NFA) program in 2003 based on Wackernagel's calculations, and, since then, the EF calculation methodology framework is regularly updated [27]. The most recent update, which contains data sets for most countries and the world from 1961 and 2017, was published in 2020 [28].

The indicator represents the size of land needed for humanity at a given level of technological development to satisfy its needs and absorb waste generated. Compared to other indicators of environmental impact, the most important advantages of the EF are the following: the EF is easy to understand, and it is relatively easy to determine the upper limit of sustainable consumption.

According to the concept of GFN, EF considers six land types: built-up land, forest products, grazing land, cropland, fishing ground, and carbon. Resource usage is expressed and measured by land usage, which are standardized with the help of equivalence factors (EQF) in global hectares (gha)—globally comparable hectares. This conversion number serves as a tool to compare different land types (e.g., cropland, forest, etc.). Since productivity of the particular land types may show regional differences, an adjustment-specific yield factor (YF) is applied [29].

Besides the spread of spatial calculations [30–34], corporate calculations were also introduced. The principles of corporate EF calculations were developed by Nicky Chambers and her colleagues in 2000 [35]. Although the concept of EF calculation was developed by examining (un)sustainability at a macro-level, it is equally useful at a micro-scale, for example, for corporations or other organizations. EF calculations could help corporations to find intervention fields [36] where environmental measures are the most effective, i.e., a particular amount of money spent has the greatest positive impact on corporate environmental performance.

A clear sign of global unsustainability of $CO_2$ emissions is that, although the usage of all land types has been increasing since the Industrial Revolution, the increase in carbon usage had the most significant role. Carbon usage grew from 43.8% to 59.9% of total land usage between 1961 and 2018, while it has an annual growth rate of 2.54%, the second highest among the land types [37].

### 2.2. Ecological Footprint as a Possible Corporate Environmental Performance Indicator

The usage of natural resources of business operations has an obvious impact. The concept of environmental performance attempts to measure and manage such impacts. Trumpp et al. [38] reviewed the related literature and identified 16 articles that give a definition of corporate environmental performance (CEP). Since 5 articles refer to the definition of International Organization for Standardization (ISO) standard 14031, and they capture the most important aspects of the 11 other definitions, the authors argue that "the ISO definition provides an encompassing and parsimonious definition". The ISO standard defines environmental performance as the "measurable results of an organization's management of its environmental aspects" [39]. However, the exact and comparable measurement of CEP is not easy because the ISO definition is "fuzzy enough to impose no clear conceptual boundaries" [40].

According to Jung et al., environmental performance measures can be grouped into five categories [41], where general environmental management (GEM) represents the strategic level, while the other four categories (input, process and operation, output, and outcome) are operational. Input measurement considers the raw material (for example, water, timber, metals, etc.) and energy (electricity, fossil fuels, etc.) consumption, while

output measures reveal desirable outputs (energy or pollutant savings) and undesirable outputs, for example, emission of air, water, or even land pollutants. As Schultze and Trommer summarize, these two measures refer to "companies' physical interactions with the natural environment" [42]. Process measures deal with optimization of corporate operations to enhance CEP, i.e., the increase in material efficiency and raising awareness of employees and suppliers. Outcome measures concern financial outcomes of the actions taken (for instance, avoided costs, fines, penalties, or even cost savings) and non-financial outcomes, which comprise mainly stakeholder relations, for example, complaints, lawsuits, or reputational issues [41].

We argue that the EF can be considered as an input/output environmental performance measure, since it focuses on the resources (raw material and energy consume, built-up land, etc.) that are consumed in business operations. Furthermore, we argue that the EF is a suitable tool to measure and manage CEP because ecological footprint:

(1) is a well-known and easy-to-understand measure of environmental sustainability;
(2) is a quantitative indicator and is measured on a ratio scale, therefore providing adequate data to create key performance indicators (KPIs);
(3) is a reliable indicator because calculations are based on scientifically proven data, such as carbon emission factors of electricity grid or fossil fuels, local food consumption, etc.; and
(4) calculations can be standardized through online calculators, therefore providing a low-cost solution for small- and medium-sized companies.

Although standardized calculations and methodologies of EF calculators can be considered as an advantage, especially for SMEs and individuals, Harangozó and Szigeti found that online corporate carbon footprint calculators may have validity and reliability issues, even in the case of the simplest business operations [15]. The authors suggested that the reliability of online EF calculators can be enhanced with more detailed input data and using local data (e.g., electricity mix). Furthermore, while corporate carbon footprint calculations are more commonly used among SMEs than EF calculations [43], understanding further aspects of EF brings new insights to improving environmental performance at the SME level.

### 2.3. Impact of Environmental Performance on Financial Performance

Although some authors suggested that the EF can be reduced at low or no cost [44], further engagements consume scarce corporate resources (e.g., financial funds, human resources, managerial attention, etc.). Since these resources could be used for other projects with net present value, companies will engage only in environmental projects the benefits of which exceed their costs. The link between sustainability and corporate financial performance (CFP) is an empirically well-studied area (see References [45–52]). Meta analyses (e.g., References [53–55]) mostly showed a positive relationship. Although there is no consensus on which indicators measure sustainability the best, we have found no study that used EF as a proxy. We suggest, however, that EF could be a suitable indicator of CEP because, (1) as we mentioned above, the EF has some advantages over other indicators; and (2) the EF is measured in ratio scale, therefore making the link between CEP and CFP examinable with more sophisticated methods than in case of other proxies measured by dummy variables (e.g., certificates, non-financial disclosures, etc.).

According to the theoretical model of Schaltegger and Synnestvedt [56], up to a point, environmental efforts pay off (see point A in Figure 1); after that, marginal benefits will be decreasing. Nonetheless, further environmental protection efforts may be confirmed because the economic performance will be higher than at the starting point up to point B in Figure 1. Two other consequences are as follows: (1) due to managerial skills, attitudes towards and the ignorance of environmental performance may vary at a given level of economic performance; and (2) several factors (e.g., change of consumer attitude, technological development, etc.) may allow to implement further environmental protection efforts, i.e., it causes the curve to shift right (see dashed line in Figure 1).

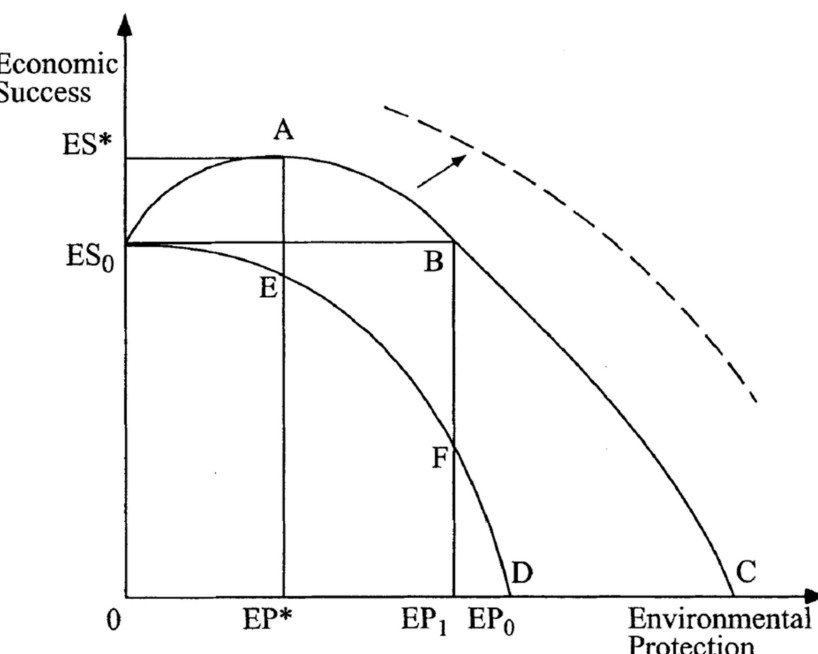

**Figure 1.** Possible relations between corporate environmental protection and economic success. Source: Reference [56].

By analyzing a sample of 4186 companies in OECD countries, empirical evidence on the positive relationship between environmental protection efforts and financial performance has been found [57]. Furthermore, Zhang et al. [58] provide a more sophisticated version of the model by adding the effects of environmental uncertainty. The authors suggest that environmental uncertainty may influence both costs and benefits of CEP through several factors. Their empirical findings show that the link between the corporate environmental performance (CEP) and the CFP is "steeper and of a lower plateau in higher levels of environmental uncertainty characterized by high dynamism, low munificence, and high complexity".

### 2.4. Sectoral Average of EF

As it was mentioned earlier, EF was developed to calculate environmental impacts of larger areas (regions, states, countries, etc.) and individuals or their households. In addition, the EF concept was complemented with other, specific calculations to determine sustainability of industrial branches or companies, among others [44]. Although ecological and carbon footprint calculations may be suitable tools for measuring both environmental and economic improvements and related reporting [36], however, one of the main limitations of corporate footprint calculations is the lack of benchmark data; namely, there are no industrial or sectoral averages available to assess the calculated footprint value. Recent research [59–62] aims to fill this gap and to provide guidance for both advisors and managers to assess CEP. To highlight both methodological approaches and impacts of different business models on EF values, in this subsection, we provide a brief insight on the results from three different specific EF calculations.

Mining is one the most $CO_2$ intensive sectors; thus, there is a legitimate demand on calculating total EF and optimizing it. Murakami et al. [59] have found that underground mines (1) have significantly lower EF for built-up land due to their smaller land-use change, and (2) fossil fuel consumption is also much lower due to their electrification; therefore, the EF could be decreased by using renewable energy sources.

Residential homes have a rather high EF in the EU. Energy consumption of households makes 26.1 percent of total final energy consumption in the EU, out of which heating is the largest portion (63.6%) [63]. Residential buildings have an average energy intensity of 180 kWh/m$^2$, but it shows significant differences among countries, even when they are

located in the same climate zone [64]. Another aspect that studies have shown is the high variability of emissions associated with construction and operation of buildings during their life cycle [65]. Since the Energy Performance of Buildings Directive requires all new buildings to be nearly zero-energy by the end of 2020 in the EU [66], EF minimalization measures should focus on the construction phase. Incorporating EF figures in construction cost databases could support in optimization of both environmental impact and costs of construction. A case study from Andalusia (Spain) highlights that the substitution of traditional construction units with lower EF solutions could result in 18% reduction of the EF, while the total cost increased only by 7% [60]. Using recycled materials (e.g., wood, concrete, steel) could reduce the EF significantly [61].

Since Hungary is an export-oriented, open, and small market economy, industrial parks can be considered as important engines of economic growth and regional development (see References [67,68]). In a case study from China [62] researchers claim that through eco-industrial transformation, EF of HETDA industry park of China can be reduced by 15.9 percent [62]. Nevertheless, other studies have shown that most eco-industrial parks are at a very early stage of development [69].

## 3. Methodology

A mixed methodology was used in this study. On the one hand, an online ecological footprint calculator was developed according to the special needs of the SME sector. A brief outline of the calculator can be found in the appendix. On the other hand, with a special regard to EF, we conducted interviews and mini case studies to gain deeper understanding of the unique features of SMEs operating in different sectors.

Both the monetary and employment figures are standardized. First, although financial data was collected in local currency (Hungarian forint, HUF) in the survey, results are expressed in euros. Since survey data considers both 2018 and 2019, an arithmetic average of daily exchange rates of the European Central Bank was applied (322.0932 HUF/EUR). Second, all employment data are expressed in full-time equivalents.

### *3.1. Calculation of EF*

The Table 1 cites only articles in which figures, methodology, etc., were directly used in the calculator developed.

Although material usage was part of a previous version of the calculator, later it was excluded from the formula due to the fact that the 500+ materials we employed in the explorative phase could not be standardized in a proper way [16].

**Table 1.** Element of ecological footprint (EF) calculated, their short description, and calculation method.

| Element of EF | Description | Calculation Method | Literature |
|---|---|---|---|
| $EF_{meals}$ | Food consumption during work time, calculated on the base of Hungarian national average values. | Equation (1) | Mózner [70] |
| $EF_{water\ consumption}$ | Water consumed by employees during work time. Industrial water consumption is excluded. | Equation (2) | Chambers et al. [35] |
| $EF_{built-up\ area}$ | Total area of non-water absorbent surfaces. | Equation (3) | Lin et al. [29] |
| $EF_{electricity\ consumption}$ | Electricity consumption from electricity grid, included heating and boiling with electric devices. | Equation (4) | IEA [71] DEFRA 2018 [72] |
| $EF_{heating\ and\ boiling}$ | Heating and boiling with fossil fuels, e.g., natural gas, coal, or wood. | Equation (5) | DEFRA 2018 [72] |
| $EF_{transportation}$ | All transportation-related EF, including commuting (both public transport and vehicles owned by employees or by the enterprise), transportation of goods, using of corporate cars, flying, etc., petrol, gasoline, and gas consumption of equipment (e.g., generators) are included. | n/a | DEFRA 2018 [72] |

The EF of meals was calculated on the basis of Hungarian average values of people's food consumption [70] (see Equation (1)). Average values do not take into consideration food consumption exceeding the minimal human needs (e.g., alcohol or candy consumption, import goods, etc.); therefore, they provide a rather lower estimate than the real figures. To achieve more accurate results, different EF factors were used for both females and males, as well as the characteristics of jobs (i.e., white collar or blue collar). Since the abovementioned values reflect the total food consumption of a given year, we assumed that employees have *n* working days a year, and they consume *i* percent of their meals at the workplace, where *n* and *i* values are given by the SMEs surveyed for each employee category. Calculation of the EF of food consumption was as follows:

$$EF_{meals} = \frac{n_{female}}{365} \times i_{female} \times \sum E_{job} \times EF\ factor_{job} + \frac{n_{male}}{365} \times i_{male} \times \sum E_{job} \times EF\ factor_{job}, \tag{1}$$

where:

$n$—number of working days of both female and male employees,
$i$—percent of at workplace consumed meals,
$E$—number of employees at a given job type (e.g., white collar or blue collar), and
*EF factor*—EF factor of each job type (e.g., white collar or blue collar).

The EF of food consumption is one of those EF elements which could differ significantly among regions [73]. An EF calculation on food consumption conducted by a Polish research team showed a much larger EF per capita figure. (It is interesting to note that Poland is another Central Eastern European country and EU member state.) The higher number is partly due to methodological considerations.

Spanish and Chilean EF values on food consumption, both of them based on Food and Agriculture Organization (FAO) of the United Nations data, show significant differences too, 0.97 and 1.43 gha per person, respectively [61].

According to our methodology, the EF of water consumption calculates with the EF of building and maintenance of water pipelines, sewage, and wastewater treating facilities. Since exact measures are not available, we assumed that the EF of water consumption is a function of employee number (see Equation (2)).

$$EF_{water} = \left(E_{female} + E_{male}\right) \times EF\ factor_{water}, \tag{2}$$

where:

$E$—number of both female and male employees; and
*EF factor*—EF factor of water consumption.

The EF of built-up area was calculated on the base of buildings' ground floor and other covered and non-water absorbent (e.g., asphalt or concrete) surface (see Equation (3)).

$$EF_{built-up} = \left(S_{building} + S_{other}\right) \times EF\ factor_{built-up}, \tag{3}$$

where:

$S$—covered surface, both ground floor of buildings and other non-water absorbent surfaces, in square meters; and
*EF factor*—EF factor of built-up area.

The EF of electricity consumption is based on carbon intensity figure (264 g $CO_2$e/kWh 2015) of International Energy Agency (IEA) [71]. $CO_2$e (carbon dioxide equivalent) is a term for describing different greenhouse gases in a common unit. For any quantity and type of greenhouse gas, $CO_2$e signifies the amount of $CO_2$ which would have the equivalent global warming impact. This value was adjusted from $CO_2$e to $CO_2$ figures by the British organization called Department for Environment, Food and Rural Affairs (DEFRA) database 2018 [72] in order to determine carbon intensity values in $CO_2$/kWh instead of in $CO_2$e. After that we added estimated impacts of energy generation and losses of electricity transmission and distribution. Although the renewable energy generation

of enterprises was taken into consideration, its EF factor was determined as 0. The EF of electricity consumption was calculated as follows:

$$EF_{electricity} = El_{grid} \times EF\ factor_{electricity} + El_{renewable\ generated} \times 0, \tag{4}$$

where:

*El*—electricity consumed (i.e., bought from the electricity grid or generated by the enterprise); and

*EF factor*—EF factor of electricity consumption.

The calculation of EF of heating and boiling is based on carbon intensity factors of DEFRA database 2018 [72]. It includes the usage of different fossil energy sources, e.g., natural gas or even burning coal.

$$EF_{heating\ and\ boiling} = \sum FES_i \times EF\ factor_i, \tag{5}$$

where:

*FES*—fossil energy source (e.g., megajoules of natural gas or tonnes of wood logs); and

*EF factor*—EF factor of specific fossil energy source.

Besides heating and boiling, transportation and the related carbon footprint generally makes up the largest portion of EF [74]; therefore, our online EF calculator provides the following options to determine the EF:

(1)   usage of different fuel types (i.e., petrol, gasoline, LPG), if accurate analytical records are available;
(2)   mileage of vehicles of different fuel types (kilometers a year) and average fuel consumption (liters per 100 km);
(3)   mileage of different category and fuel type of cars and small vans;
(4)   number and average distance of trips in case of taxi and air travel; and
(5)   an average of daily distance in case of public transport (underground, tram, bus).

Since SMEs in general use several different transportation modes, only the first two calculation methods are mutually exclusive. All carbon intensity factors are based on the DEFRA 2018 database [72].

*3.2. The Sample*

Enterprises in our sample were required to have the following attributes:

(1)   It is a small- or medium-sized company, defined by the Commission of the European Communities [75], namely has less than 250 employees and its turnover is less than €50 million or its balance sheet total is less than €43 million.
(2)   Energy consumption of corporate activities can be separated from other activities, e.g., private home of managers and/or owners.
(3)   Managers and/or owners are willing to participate in the survey.

Data was collected from three sources: (1) SMEs known from our professional network or from our university networks; (2) commercial and industrial chambers in Hungary were asked to send calls for survey to their member companies, and we participated in some of their events; and (3) students were asked to assist with our study. Mini case studies were conducted about most of the companies surveyed to gather additional qualitative data.

Companies were filtered out from our sample as an outlier when one or more figures varied significantly from other companies of the same group and we had no plausible explanation for this (e.g., equipment used, working processes, etc.).

Anecdotal evidence suggest that the SMEs of different business activities may have similar EF. Therefore, a preliminary qualitative analysis was conducted to classify SMEs on the basis of the determining factors of their ecological footprint, i.e., based on the attributes of their CEF. This is inevitably different from statistical classifications (i.e., NACE in the EU or SIC in the USA). We suggest that a more detailed and more accurate result could be achieved by analyzing a larger database. For example, white-collar jobs have similar

environmental impact, regardless of whether the enterprise is involved in bookkeeping, software development, civil engineering planning, or even fashion design. The ecological footprint of white-collar jobs is determined mainly by (1) the conditions of the property used (place, size, insulation, effectiveness, and usage of air conditioning and heating, etc.), (2) commuting habits of employees and home office opportunities, (3) number and length of business trips and vehicles used, and (4) the number of employees.

The study focuses on the following five groups of SMEs (see Table 2):

**Table 2.** Classification of SMEs analyzed.

| Name of Group | Common Sense | Related Subsection |
|---|---|---|
| construction | Extensive use of machines, heavy-duty vehicles. EF is determined mostly by fossil fuel consumption. | Section 4.1 |
| white-collar jobs | Knowledge-intensive activities, moderate land use, equipment with low consumption (e.g., laptops, plotters, etc.). Vehicle usage is limited for passenger cars and only for field visits or commuting. EF is rather balanced among determining factors. | Section 4.2 |
| production | Technology-intensive activities, significant usage of equipment and land. EF is determined mostly by energy and fossil fuel consumption, but built-up land usage and food consumption are also significant. | Section 4.3 |
| retail and/or wholesale trade | Significant land use (buildings and parking lots), moderate use of equipment (e.g., refrigerators). Moderate vehicle usage. EF is determined significantly by heating and boiling; fuel consumption could be significant in case of home delivery or other vehicle usage. | Section 4.4 |
| transportation | Extensive use of trucks and other resource usage is negligible. EF is determined most of all by gasoline consumption. | Section 4.5 |

Variation of EF among group of enterprises can be explained by several coexisting factors:

(1)　The operation of SMEs may differ. For example, the EF will be greater if a retail store transports goods with its own van and/or provides home delivery for costumers, or if an engineering office must make trips for its field works.

(2)　Manager's attitudes towards sustainability may vary significantly. Some managers attempt to engage in environmentally friendly projects (e.g., energy efficient equipment, solar panels, etc.), while others do not.

(3)　The organization culture may also be different.

One of the limitations of the EF calculator is that it ignores all the factors that are beyond the control of companies. Accordingly, financial performance is measured by an adjusted value added, which is calculated on the available accounting data as the sum of personnel costs, amortization, and after-tax profit. Adjustment had to be made because of a simplified tax type eligible only for small companies. If a company chooses this tax type, it substitutes corporate tax and social contributions of employment. Since personnel costs of companies of different types are directly not comparable, we chose after-tax profit instead of pre-tax profit. We suggest that these kinds of calculations provide more comparable results among the analyzed SMEs but have the limitation that all value-added figures presented show an underestimation of real values.

## 4. Results

Our sample consists of 73 SMEs from the five groups. Four out of the five groups have 15–20 valid items, while the smallest sub-sample (transportation) comprises only 4 items. This can be explained by the relative simpleness of the sector; the EF of these SMEs is determined almost completely by fuel consumption (liters of diesel per 100 km). Detailed results are presented in the following subsections. For detailed numerical information see Tables 3 and 4.

**Table 3.** Descriptive statistics.

| | | | Construction | White-Collar Jobs | Production | Retail and Wholesale Trade | Transportation |
|---|---|---|---|---|---|---|---|
| | Valid cases | | 17 | 17 | 15 | 20 | 4 |
| specific EF (global hectares/employee) | Mean | | 1.25 | 0.46 | 1.47 | 1.10 | 20.15 |
| | 95% Confidence Interval for Mean | Lower Bound | 0.87 | 0.32 | 0.85 | 0.73 | 17.00 |
| | | Upper Bound | 1.62 | 0.60 | 2.08 | 1.47 | 23.30 |
| | 5% Trimmed Mean | | 1.20 | 0.43 | 1.42 | 1.06 | 20.20 |
| | Median | | 0.93 | 0.44 | 1.21 | 0.81 | 20.56 |
| | Std. Deviation | | 0.72 | 0.27 | 1.11 | 0.79 | 1.98 |
| eco-efficiency (global hectares/th. EUR) | Mean | | 0.089 | 0.051 | 0.067 | 0.088 | 1.055 |
| | 95% Confidence Interval for Mean | Lower Bound | 0.065 | 0.029 | 0.033 | 0.050 | 0.410 |
| | | Upper Bound | 0.113 | 0.074 | 0.100 | 0.126 | 1.701 |
| | 5% Trimmed Mean | | 0.086 | 0.047 | 0.064 | 0.079 | 1.040 |
| | Median | | 0.076 | 0.041 | 0.047 | 0.071 | 0.918 |
| | Std. Deviation | | 0.047 | 0.043 | 0.061 | 0.081 | 0.406 |
| specific value added (th EUR/employee) | Mean | | 15.40 | 15.29 | 32.98 | 17.24 | 20.64 |
| | 95% Confidence Interval for Mean | Lower Bound | 11.94 | 7.23 | 14.04 | 12.64 | 11.79 |
| | | Upper Bound | 18.85 | 23.34 | 51.93 | 21.84 | 29.49 |
| | 5% Trimmed Mean | | 14.99 | 13.01 | 27.78 | 16.83 | 20.74 |
| | Median | | 14.96 | 11.38 | 22.97 | 15.78 | 21.57 |
| | Std. Deviation | | 6.71 | 15.67 | 34.21 | 9.83 | 5.56 |

**Table 4.** Correlations.

| Activity | | | Specific EF (gha/empl) | Eco-Efficiency (gha/th EUR) | Specific Value Added (th EUR/empl) | Activity | Specific EF (gha/empl) | Eco-Efficiency (gha/th EUR) | Specific Value Added (th EUR/empl) | Activity | Specific EF (gha/empl) | Eco-Efficiency (gha/th EUR) | Specific Value Added (th EUR/empl) |
|---|---|---|---|---|---|---|---|---|---|---|---|---|---|
| construction | specific EF (gha/empl) | Pearson Correlation | 1 | 0.778 ** | 0.177 | white-collar jobs | 1 | 0.524 * | −0.042 | production | 1 | 0.788 ** | −0.209 |
| | | Sig. (2-tailed) | | 0.000 | 0.497 | | | 0.037 | 0.878 | | | 0.001 | 0.472 |
| | | N | 17 | 17 | 17 | | 16 | 16 | 16 | | 14 | 14 | 14 |
| | eco-efficiency (gha/th EUR) | Pearson Correlation | 0.778 ** | 1 | −0.408 | | 0.524 * | 1 | −0.507 * | | 0.788 ** | 1 | −0.378 |
| | | Sig. (2-tailed) | 0.000 | | 0.104 | | 0.037 | | 0.045 | | 0.001 | | 0.182 |
| | | N | 17 | 17 | 17 | | 16 | 16 | 16 | | 14 | 14 | 14 |
| | specific value added (th EUR/empl) | Pearson Correlation | 0.177 | −0.408 | 1 | | −0.042 | −0.507 * | 1 | | −0.209 | −0.378 | 1 |
| | | Sig. (2-tailed) | 0.497 | 0.104 | | | 0.878 | 0.045 | | | 0.472 | 0.182 | |
| | | N | 17 | 17 | 17 | | 16 | 16 | 16 | | 14 | 14 | 14 |
| retail and wholesale trade | specific EF (gha/empl) | Pearson Correlation | 1 | 0.379 | 0.245 | transportation | 1 | 0.591 | −0.406 | | | | |
| | | Sig. (2-tailed) | | 0.099 | 0.298 | | | 0.409 | 0.594 | | | | |
| | | N | 20 | 20 | 20 | | 4 | 4 | 4 | | | | |
| | eco-efficiency (gha/th EUR) | Pearson Correlation | 0.379 | 1 | −0.543 * | | 0.591 | 1 | −0.960 * | | | | |
| | | Sig. (2-tailed) | 0.099 | | 0.013 | | 0.409 | | 0.040 | | | | |
| | | N | 20 | 20 | 20 | | 4 | 4 | 4 | | | | |
| | specific value added (th EUR/empl) | Pearson Correlation | 0.245 | −0.543 * | 1 | | −0.406 | −0.960 * | 1 | | | | |
| | | Sig. (2-tailed) | 0.298 | 0.013 | | | 0.594 | 0.040 | | | | | |
| | | N | 20 | 20 | 20 | | 4 | 4 | 4 | | | | |

**. Correlation is significant at the 0.01 level (2-tailed). *. Correlation is significant at the 0.05 level (2-tailed).

### 4.1. Construction

Activities of construction enterprises in our sample, ranging from civil engineering, structural architecture, and some special construction firms (e.g., planning, implementing solar panels and other electric equipment on buildings, installing shading equipment, etc.), are also present. They have an average EF of 1.25 gha/employee (confidence interval (CI): 0.87–1.62), an eco-efficiency of 0.089 gha/thousand EUR adjusted value added (CI: 0.065–0.113), and specific value added of 15.4 thousand EUR/employee (CI: 11.94–18.85). Positive correlation between eco-efficiency and specific EF ($p < 0.01$) shows that more eco-efficient construction also has lower the EF per employee figures. Significant correlations between other variables could not be identified.

The EF of construction enterprises is determined mostly by the consumption and efficiency of vehicles and other equipment used. Our mini cases show that managers mostly aimed to reduce fuel consumption; therefore, vehicles are regularly replaced by more efficient ones, private vehicle use is restricted, and employees are collected by a company vehicle. It is interesting to note, however, that the prestige of driving a car is of great importance for many people, and they drive to work even if the commuting distance is less than a few kilometers. Nevertheless, a moderate vehicle use may be allowed in most construction enterprises, since the second half of 2010s is marked with a shortage of trained and experienced professionals. Another issue is that, although there is governmental aid for purchasing battery electric vans or cars, the managers interviewed are concerned about the higher price and the lack of experience; therefore, only a small car that was used for the everyday corporate errands was to be replaced.

If the company has a larger office building, it is often retrofitted or is even equipped with solar panels.

### 4.2. White-Collar Jobs

White-collar jobs include mostly financial and accounting services (bookkeeping, tax advisory services, auditing, etc.), but engineering, education, or even software development enterprises are present in the sample. The group has the smallest environmental impact—an average of 0.46 gha/employee (CI: 0.32–0.60) and average eco-efficiency of 0.051 gha/thousand EUR adjusted value added (CI: 0.029–0.074), while the average specific value added is less than in other sectors, 15.29 thousand EUR/employee (CI: 7.23–23.34). Results of the correlation analysis show that (1) more eco-efficient enterprises have significantly lower specific EF figures ($p < 0.05$) and (2) higher specific value added ($p < 0.05$). This latter result means that engagement in environmental protection measures and/or project may be profitable.

Since working in an office is a human capital-intensive activity, its EF is determined mostly by the energy-efficiency of the buildings used and by the commuting practices and working trips of the workforce. While the former figure can easily be reduced by insulation and/or renovation of the buildings, by using energy-efficient lightning or even by implementing solar panels, reducing the latter figure is a more complicated issue. On the one hand, the COVID-19 pandemic showed that personal contacts can be at least partly substituted by online meetings, but working trips could be necessary in some cases; for example, engineers must visit working fields or even cultural determinations may require personal meetings. On the other hand, the prestige of commuting by car and/or living in urban agglomerations may influence the habits of employees. Furthermore, employees mostly use their own cars; therefore, it is out of the managers' control. Based on our findings, we recommend promoting more sustainable ways of commuting. For example, when it is feasible, businesses should provide shower and changing facilities for cyclists in the workplace, but biking events and/or actions may influence commuting habits, as well. Of course, financial stimuli could also be used, for example, cutting contributions on commuting with a car and providing benefits for public transport usage instead.

### 4.3. Production

Producer companies in our sample are very diverse—they range from manufacturing spices, wooden toys for playgrounds to producing vehicles. The EF figures of these activities differ substantially. Specific EF is EF 1.47 gha/employee (CI: 0.85–2.08) on average in this group, while eco-efficiency is favorable, 0.067 gha/thousand EUR (CI: 0.033–0.100), and specific value added is 32.98 thousand EUR/employee (CI: 14.04–51.93) due to the higher adjusted value added. Just as in the case of construction enterprises, correlation analysis shows a significant relationship only between eco-efficiency and specific EF ($p < 0.01$).

Production is technology-intensive, so EF is also highly determined by working processes and equipment used. Our mini cases show that companies attempt to implement both up-to-date working processes and efficient equipment, but EF figures are influenced significantly by other factors, such as industrial specialties, level of market competition, managerial attitudes, and governmental and/or EU grants.

### 4.4. Retail and Wholesale Trade

Retail and/or wholesale trade companies range from pharmacies and other fast-moving consumer goods (FMCG) stores to wholesale of electronic components or even veterinary items. The most significant difference among companies is the following: (1) transportation and/or home delivery of goods with own vehicle or by a third party; and (2) special storage needs of goods sold (e.g., storage of frozen or chilled goods have much higher energy consumption than of recyclable waste). Specific EF of the sector is on average 1.10 gha/employee (CI: 0.73–1.47), and eco-efficiency is 0.088 gha/thousand EUR (CI: 0.050–0.126), while specific added value lies at 17.24 thousand EUR/employee (CI: 12.64–21.84). We found significant correlation between eco-efficiency and specific value added ($p < 0.05$). It means, as in the case of office activities, that more eco-efficient enterprises have generally higher added value per employee; namely, there is a positive relationship between corporate environmental performance and value-added creation.

Our cases reveal that companies of the clusters sector have similar challenges as of offices, namely energetical characteristics of the buildings used.

### 4.5. Transportation

The fifth group is transportation, which is the most EF-intensive sector in our analysis. Specific EF figure of transportation companies is 20.15 gha/employee (CI: 17.00–23.30), which is 16 times higher than of construction companies. Average eco-efficiency is 1.055 gha/thousand EUR (CI: 0.410–1.701), and specific value added is 20.64 thousand EUR/employee (CI: 11.79–29.49). Similar to other groups, our correlation analysis identifies significant relationship between eco-efficiency and specific value added ($p < 0.05$), establishing a positive connection between corporate financial and environmental performance.

Our cases show that there are four main routes to reducing the EF: (1) increasing the efficiency of vehicle technology, which means not only lower consumption in relative terms (liters per 100 km), but highway tolls and maintenance costs are significantly lower, as well; (2) monitoring fuel consumption could mitigate misuse of tanked fuel and provide data for route optimization; (3) route optimization could decrease mileage of trucks, which means lower consumption in absolute terms (liters per trip); and (4) using lower-carbon fuels (e.g., hydrogen).

## 5. Conclusions and Discussions

The paper aimed to develop an easy-to-use EF calculator for SMEs which could measure the common elements of corporate environmental impacts reliably. Results are based on a sample of 73 corporate EF calculated by an online EF calculator; thus, identical approach and methodology was assured.

Our results primarily have practical implications, as they show that it is feasible to develop an EF calculator for SMEs which can provide reliable figures and is easy-to-use. As anecdotal evidence suggested, SMEs can be classified on the basis of the

determining factors of their ecological footprint. Considering their EF figure calculated with a standardized methodology, benchmark data could be also calculated to measure CEP. Using a larger sample, a more detailed classification and more accurate benchmark data could be provided.

According to our results, there is significant and negative correlation between eco-efficiency (EF/value added) and added value per capita in some groups (office activities, retail and/or wholesale trade, transportation). Similarly, significant correlation cannot be found in other analyzed groups (construction, production). These findings suggest that CEP does not influence the financial performance of the analyzed SMEs negatively; rather, there is a positive link in the case of some groups. A possible explanation for the difference of sectors' results may be that production and construction are both highly technology intensive sectors; therefore, environmental protection measures are either too expensive (e.g., more advanced production technology) or there is no available solution (e.g., heavy duty vehicles with electric powertrain).

It is remarkable that transportation enterprises have much higher EF figures than enterprises from other groups. On the one hand, it highlights the importance of locality [76]. On the other hand, transportation connect participants of the value chains. It means that activities with significantly different EF figures in a value chain are separated into several enterprises, so it would seem that CEP of a specific enterprise might be high, but actually only a more environmentally intensive element of the value chain is outsourced to it.

Our results have four main limitations. First, there is no widely used and accepted methodology of conducting easy-to-use and reliable EF calculator for SMEs; therefore, we could not lean on former experiences or calculators that we could have used for the development and testing of our calculator. To provide reliable results, only the common elements of corporate EF were taken into consideration. Although EF of material usage might make up a significant proportion of corporate EF, we suggest that the number and diversity of materials would make the calculator too complex and complicated to use. Second, the sample used in the analysis is small and does not represent the real environmental performance of Hungarian SMEs. Furthermore, we suggest that the sample is positively biased because companies with higher environmental performance are more willing to participate in the survey. Third, although the most financial data were validated on the basis of disclosed financial reports, and we adjusted them according to findings of qualitative methods, firm-specific parameters (e.g., part-time employment, tax optimization, accounting policies, business models, etc.) could significantly influence the results. Fourth, our calculator does not consider material usage of companies; thus, the provided EF values are consistently underestimated.

The results presented in this article show a transition phase between individual and mass calculations; therefore, our future research aims to provide more accurate benchmark data on EF values of SMEs based on a larger sample size. This step would make it possible to conduct more sophisticated analyses using moderating variables (such as corporate governance [77,78], family businesses [79], developed, emerging, and transitional countries, etc.). Another aspect could be to complement the data set with other sectors, for example, with services, agriculture, etc., and to compare EF values of SMEs based in different countries.

**Author Contributions:** Conceptualization and methodology, C.S.; data collection, Á.S., J.B., C.S.; validation and formal analysis, Á.S.; writing—original draft preparation, Á.S.; writing—review and editing, J.B., L.R.; supervision and project administration, C.S., L.R. All authors have read and agreed to the published version of the manuscript.

**Funding:** This research was supported by a grant from the Higher Education Institutional Excellence Program of the Hungarian Ministry of Innovation and Technology to Budapest Business School (NKFIH-1259-8/2019). The APC was funded by Budapest Business School—University of Applied Sciences.

**Institutional Review Board Statement:** Not applicable.

**Informed Consent Statement:** Not applicable.

**Data Availability Statement:** The data are not publicly available because respondents did not permit data usage of third parties.

**Conflicts of Interest:** The authors declare no conflict of interest.

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
