# Peer review of "Ecological Footprint as an Indicator of Corporate Environmental Performance—Empirical Evidence from Hungarian SMEs"

_sustainability, doi:10.3390/su13021000_

Round 1
Reviewer 1 Report
The Abstract must have the following logic: Purpose; Design / methodology / approach; Findings; Practical implications; Originality / value
70 or 73?
Introduction - The GAP must be highlighted. The purpose of the study is not clear.
The point “The concept of ecological footprint” needs to be improved. For example: What is the evolution of the concept?
Regarding the indicators, some studies use the same indicators and are not in the literature review.
They could make a table with the definition of each indicator and the authors who studied them.
Some relevant studies:
- National natural capital accounting with the ecological footprint concept
- Perceptual and structural barriers to investing in natural capital: Economics from an ecological footprint perspective
- The Review of Footprint analysis tools for monitoring impacts on sustainability
- Integrating Ecological, Carbon and Water footprint into a "Footprint Family" of indicators: Definition and role in tracking human pressure on the planet
- Establishing national natural capital accounts based on detailed - Ecological Footprint and biological capacity assessments
- Spatial sustainability, trade, and indicators: an evaluation of the 'ecological footprint'
- Research on the ecological footprint of tourism: the case of Langzhong in China
- The interplay among ecological footprint, real income, energy consumption, and trade openness in 13 Asian countries
- Global ecological footprint and spatial dependence between countries Manuel
- Determinants of the ecological footprint: Role of renewable energy, natural resources, and urbanization
The methodology needs to be justified. Why did you use this methodology and not another?
In table 1, do you only have one author for “Element of EF”? There are other authors (which should be addressed in the literature review.)
The sample of 73 SMEs is very short.
Statistically, the study is not robust. In general, you must have more than 200 responses to be able to make a statistically robust study.
The discussion of the results does not exist. As the statistics are not robust, the conclusions are also not significant. What is the originality of the study? What are the theoretical and practical implications?
Author Response
Thank you for your valuable suggestions and comments. Please, see the attachment.

Reviewer 2 Report
Dear authors,
thank you - very interesting reading,
i have same notes:
- the presented study represents a well-processed and summarized set of information about the new method of assessing companies, ie about the ecological footprint and its implementation into the environment.
- the methodology of the article describes the procedure of the authors well, or is clear
- proposal - in the abstract to emphasize more the benefit of this methodology in terms of how it can be used for society
Author Response

(The authors gave the same response as above.)

Reviewer 3 Report
-A very interesting topic with some very good empirical analysis.
-The paper is well structured.
-The author correctly used the existing literature .
-Appropriate research methods have been selected, applied with care, rigour and careful planning.
- The results or outcomes are accurately presented, interpreted -
-I strongly suggest that the author should extend the conclusion, especially second paragraph , and discuss more what those results mean.
-Author suggests some further improvements, however, I would recommend to include also the limitations of this study. If those will be taken into account in future. Also, a similar examination of all EU countries will be good for future research
-Another suggestion-not essential is in section 4 to write a short introduction and explain what you will have in this section.
-Finally, I would change the expression "our professional experience" in section 1.
Author Response

(The authors gave the same response as above.)

Reviewer 4 Report
This paper is focused on the use ecological footprint for environmental as an environmental performance indicator for SMEs. The topic fits the aims of journal but the authors should do some changes before its publication.
I would like to highlight the following points of the article that I think should be revised:
1) Introduction
The objectives of the paper should be clearly defined in this section, that is what are the research questions.
My suggestion is that the authors use the introduction section for: a) establish the context of the work being reported, summarizing the current understanding of the problem and why is interesting to study it; b) state the purpose of the work that now is not clearly defined; c) a short reference to the method used and the main results.
2) Theoretical framework
I think that the theoretical framework should be better connected with the aims of the paper.
For example, the authors dedicate a section to the concept of ecological footprint, which I really do not consider necessary for a scientific journal. They can just summarize the ideas inside the next section.
In the second section, first the authors define CEF, which again is well recognized in the literature and then they argue that the EF is a suitable tool to measure and manage CEP. However, this would require some references or previous studies that evidence it. Otherwise, the authors should show further evidence of all the benefits that they argue and how they support the suitability.
Sections 2.3 and 2.4 also should be more connected with the aim of the paper
3) Methodology
It seems that the main contribution of the paper is development of a EF calculator, included in this section. I think that this requires also some evidence or previous support that gives reliability to the indicator, in particular to the elements. The use of previous literature about it can be very useful.
4) Results
This section describes the results of the EF calculator for the different sectors, but I really find it scarce. In my opinion, in order to give value to the results, they should try to analyze the implications of the results. What is really the test to evidence the acceptance of EF as environmental indicator for SMEs?
5) Conclusion and discussion
The section is really poor and does not allow to value the contribution of the paper
In summary, I think that the paper needs some changes before its publication in Sustainability.
Author Response

(The authors gave the same response as above.)

Reviewer 5 Report
Generally, this paper do not contribute to the literature on corporate environmental performance. With poor literature review, including only few papers on the subject, the authors do not succeed to convince me that the subject worth investigation.
It is not clear what the objective of this study is and how this paper will contribute to the literature.
The authors claim that the ecological footprint is easy to understand and to use to determine the upper limit of sustainable consumption, without providing an example of ecological footprint at corporate level. The definition of the concept is poor and no definition of ecological footprint at corporate level was included in the paper.
Environmental performance is a multidimensional construct. Extensive literature exist on environmental performance. The authors limited their ‘theoretical framework’ only to few papers, relevant papers on the subject are omitted.
It is not clear why the impact of environmental performance on financial performance is described.
I have serious problems considering the calculation of corporate environmental footprint, as long as an important factor such as material consumption is not taken into account.
Author Response

(The authors gave the same response as above.)

Round 2
Reviewer 1 Report
Accept in present form
Author Response
Thanks for your suggestions and feedback!
Reviewer 4 Report
Dear authors,
Thanks for the changes.
Author Response

(The authors gave the same response as above.)
